# Leveraging Machine Learning for Severity Level-Wise Biomarker Identification in Prostate Cancer Microarray Gene Expression Data

**DOI:** 10.3390/biomedicines13102350

**Published:** 2025-09-25

**Authors:** Ahmed Al Marouf, Tarek A. Bismar, Sunita Ghosh, Jon G. Rokne, Reda Alhajj

**Affiliations:** 1Department of Computer Science, University of Calgary, Calgary, AB T2N 1N4, Canada; rokne@ucalgary.ca (J.G.R.); alhajj@ucalgary.ca (R.A.); 2Department of Pathology and Laboratory Medicine, Cumming School of Medicine, University of Calgary, Calgary, AB T2N 1N4, Canada; tabismar@ucalgary.ca; 3Departments of Oncology, Biochemistry and Molecular Biology, Cumming School of Medicine, Calgary, AB T2N 1N4, Canada; 4Arnie Charbonneau Cancer Institute, Tom Baker Cancer Center, Calgary, AB T2N 1N4, Canada; 5Alberta Precision Laboratories, Calgary, AB T2V 1P9, Canada; 6Prostate Cancer Center, Calgary, AB T2V 1P9, Canada; 7Department of Oncology, University of Alberta, Edmonton, AB T6G 1Z2, Canada; sunita.ghosh@ualberta.ca; 8Department of Computer Engineering, Istanbul Medipol University, 34810 Istanbul, Turkey; 9Department of Health Informatics, University of Southern Denmark, 5230 Odense, Denmark

**Keywords:** prostate cancer, biomarker identification, tissue microarray, XGBoost

## Abstract

**Background:** Prostate cancer is the most commonly occurring cancer amongst men. The detection and treatment of this cancer is therefore of great importance. The severity level of this cancer, which is established as a score in the Gleason Grading Group (GGC), guides the treatment of the cancer. **Methods:** In this paper, traditional machine learning (ML) classification methods such as Decision Tree (DT), Random Forest (RF), Support Vector Machine (SVM), and XGBoost (XGB), which have recently been shown to accurately identifying biomarkers for computational biology, are leveraged to find potential biomarkers for the different GGC scores. A ML framework that maps the Gleason Grading Group (GGG) into five severity levels—low, intermediate-low, intermediate, intermediate-high, and high—has been developed using the above methods. The microarray data for this ML method have been derived from immunohistochemical tests. The study includes severity level-wise biomarker identification, incorporating missing value imputation, class imbalance handling using the SMOTE-Tomek link method, and stratified k-fold validation to ensure robust biomarker selection. **Results:** The framework is evaluated on prostate cancer tissue microarray gene expression data from 1119 samples. A combination of high-aggressive and low-aggressive signatures are used in four experimental setups. The results demonstrate the effectiveness of the approach in distinguishing between critical biomarkers with highly accurate models, obtaining 96.85% accuracy using the XGBoost method. **Conclusions:** Leveraging ML gives a potential ground to involve the domain experts and the satisfactory results have approved that. For the future physician-in-the-loop approach can be tested to ensure further diagnosis impact.

## 1. Introduction

The detection and stratification of severity levels for prostate cancer is of great importance due to the disease’s heterogeneity and varied clinical courses since this stratification can optimize the treatment strategies. In particular, the accurate assessment of the severity of the cancer provides a clinician with critical information as to whether or not surgical intervention should be performed. This can then minimize overtreatment in less severe low-risk forms of the cancer while ensuring more appropriate treatments for more aggressive high-risk forms of the cancer. Hence, together with early detection, the accurate differentiation between low-risk and high-risk prostate cancer cases means that unnecessary invasive procedures for indolent tumors can be avoided while prioritizing aggressive therapies for potentially life-threatening malignancies.

The stratification of prostate cancer (PCa) severity levels thus provides personalized patient care, optimizes outcomes, and minimizes the adverse effects associated with treatments. The paper by Cooperberg et al. [1] provided a further discussion of the importance of risk stratification in prostate cancer management and the role of risk assessment tools in guiding treatment decisions and improving patient outcomes. The paper also emphasizes that the identification of high-risk cases enables timely interventions, potentially curbing disease progression and enhancing long-term survival rates. A similar discussion is found in the work of Loeb et al. [2], which highlights the importance of risk stratification in reducing prostate cancer mortality by targeting aggressive tumors. This risk stratification approach is further reinforced by the work of Dahabreh et al. [3], where the importance of risk-based treatment strategies in prostate cancer treatment for improving clinical outcomes while reducing unnecessary interventions is similarly discussed.

Hence, the accurate detection and stratification of prostate cancer severity levels are fundamental when tailoring treatment approaches, optimizing patient outcomes, and mitigating the risks associated with both overtreatment and undertreatment. This stratification aligns with the paradigm of precision medicine, ensuring that interventions are tailored to individual patients, based on the specific characteristics and the level of aggressiveness of their disease.

An analysis on a comprehensive clinically informed map of dependencies on different cancer types has been presented by Pacini et. al. [4]. They produced a second-generation map considering 27 cancer types, extracting 370 anti-cancer priority targets. This map illustrates network-based evidence for a functional link with a marker in a type of cancer. Another analysis of a transformer-powered graph representation learning-based method has been present by Su et. al. [5], where they considered cancer genes in biological networks. The interactions between miRNA and proteins, transcription factors and proteins, and transcription factors and miRNA were analyzed. Using pan-cancer analysis [6] 57 candidates were identified from the cancer genes.

Several studies have explored automated methods for predicting Gleason scores and grading groups using diverse data modalities, including histopathological images, radiomics, and genomic signatures. Table 1 provides a comparative summary of representative approaches, highlighting their input modalities, methodological strategies, reported performance, strengths, and key limitations. This overview helps us to contextualize our work and underscores the gap we address by focusing on severity level-wise biomarker discovery from prostate cancer tissue microarray gene expression data. Most prior work on Gleason grading focuses on image-based deep learning from biopsy/WSI or radiomics from mpMRI, with limited emphasis on severity level-wise biomarker discovery and scarce integration of class imbalance-aware pipelines. Existing high-performing systems are often black-box in nature, lack reproducible biomarker sets, and rarely subdivide the intermediate group into clinically relevant strata (intermediate-low vs. intermediate-high).

Although the machine learning methods employed (DT, RF, SVM, and XGBoost) are standard, the novelty of this study lies in the design of a severity level-specific biomarker identification framework, incorporating robust pre-processing, class imbalance correction, and independent validation. This approach addresses the gaps in the prior literature, which often limited analyses to binary Gleason groupings or lacked external validation.

In this paper, the focus is on the identification and stratification of the multi-label severity levels or risk levels of prostate cancer from biopsy analyses. These risk levels are determined based on the Gleason Grade Group scores, hereafter referred to simply as Gleason scores. To find the best characteristics or gene signatures responsible for each of the risk levels, machine learning (ML) methods are utilized to provide multiclass classifications. The methods selected have been shown to provide good classifications and predictions for such tasks. The main contributions of this research are the following:We proposed a machine learning framework that maps Gleason Grading Groups (GGGs) into five clinically meaningful severity levels (low, intermediate-low, intermediate, intermediate-high, and high), enabling fine-grained risk stratification in prostate cancer.The study integrates advanced pre-processing techniques—including missing value imputation, class imbalance correction using SMOTE-Tomek links, and stratified k-fold cross-validation—to ensure reliable and reproducible biomarker discovery from prostate cancer tissue microarray data.By leveraging multiple ML methods (DT, RF, SVM, and XGBoost), our framework achieves state-of-the-art performance, with XGBoost attaining 96.85% accuracy, demonstrating the potential for clinical translation in diagnosis and treatment guidance.

A synopsis of the severity levels of prostate cancer based on GGG is first discussed in the next subsection. The methods used in this research are then described in the next section. A more detailed description of the severity levels defined for prostate cancer are presented in the following section. The novel dataset used for this investigation and the data preparation steps are then described in detail. A comprehensive result section shows all the experimental findings from each step, followed by a conclusion summarizing the overall outcome of the research.

### Prostate Cancer Severity Levels—An Overview

Most prostate cancers are classified as adenocarcinomas and their severity risk levels are classified using the Gleason score using the Gleason Grading System developed by Dr. Donald Gleason in the 1960s [14,15]. The Gleason score offers pathological prognostic insight through the evaluation of the microscopic structure of cancer cells together with the clinical staging of the disease. The evaluation of how healthy or abnormal the cancer cells appear under a microscope provides a measure of the cancer’s aggressiveness. This is carried out by examining prostate tissue samples obtained by biopsy or from the collection of small tissue samples during surgery. The samples are then examined by experts (pathologists) using a microscope to find patterns in the appearance of the cells. The organization of cancer cells within the prostate is then assessed, and the pathologist assigns a primary Gleason grade between 1 and 5 to the primary prevalent pattern observed during the biopsy. Similarly, a secondary Gleason grade, also between 1 and 5, is given to the second most prevalent pattern. These grades are combined to calculate the total Gleason score on a scale from 2 to 10. However, following Dr. Gleason’s original classification, pathologists rarely assign scores below 6. Consequently, scores generally fall between 6 and 10.

The mapping of severity levels and Gleason grading groups is presented in Table 2. Risk levels, low, intermediate, or high, are determined on the basis of these scores as follows: A low-risk level constitutes a total Gleason score equal to or below 6. A Gleason score of 7, regardless of the combination of major and minor Gleason grades (3 + 4 or 4 + 3), is classified as intermediate risk, while a Gleason score 8 is classified as intermediate/high-risk. Gleason score 9 (4 = 5 or 5 + 4) and Gleason score 10 are classified as high-risk.

The Gleason grading system is the most widely used histopathological tool for prostate cancer prognosis. Its clinical significance lies in its ability to stratify patients into distinct risk categories that guide treatment decisions and predict outcomes.

Low-risk (≤6): These tumors are generally well-differentiated, less aggressive, and associated with favorable prognosis. Many patients may be candidates for active surveillance rather than immediate intervention. Intermediate risk with Gleason Score 7 is divided into two risk groups, Intermediate-Low and Intermediate-High.

Gleason 3 + 4 = 7 (Intermediate-Low) indicates that the predominant tumor pattern is less aggressive (grade 3), with only a minor component of grade 4. These patients usually have better outcomes and may be managed with less intensive therapy compared to higher-risk groups. Gleason 4 + 3 = 7 (Intermediate-High) reflects a predominance of grade 4 (more aggressive) tumor pattern. These patients have worse prognoses compared to 3 + 4 and often require more aggressive treatment. Distinguishing between “intermediate-low” (3 + 4) and “intermediate-high” (4 + 3) is clinically important because it directly influences prognosis and recommended therapeutic strategies.

Gleason 8 (Intermediate/High-risk) tumors at this stage exhibit poor differentiation and higher metastatic potential, typically prompting combined treatment strategies (e.g., surgery plus radiation and/or androgen deprivation therapy). High-risk (≥9) Gleason 9–10 cancers are very aggressive, strongly associated with early recurrence, metastasis, and cancer-specific mortality. These patients usually require multimodal and intensive treatment regimens. This risk stratification complies with clinical guidelines (e.g., NCCN [16], AUA [17,18], EAU [19]) and is essential for individualized patient management.

## 2. Methods

The framework proposed to identify biomarkers by severity level is now described in detail. The overall methodology has two main steps, namely, data preparation and machine learning.

The data preparation has several steps, such as data formatting, cleaning, merging with clinical data, handling missing values, and the class imbalance problem, using a formalized method. These steps are discussed in the following subsections.

The proposed methodology is presented in Figure 1, showing all the steps needed to identify the severity level-wise biomarkers.

### 2.1. Proposed Methodology

A machine learning (ML) pipeline was developed, as shown in Figure 1. The performance of each ML model applied in the pipeline was evaluated through a sequential process that includes data pre-processing, model training, validation, testing, and final quantification. These steps are described in detail in the following subsections. The comparative analysis of different ML approaches is quantified by the resulting performance metrics. The following machine learning algorithms were used: Decision Tree (DT), Support Vector Machine (SVM), and ensemble techniques like Random Forest (RF) and XGBoost (XGB), as well as other ML tools.

Decision Trees (DTs) [20] are highly valued for their simplicity and ease of interpretation. Furthermore, users can follow the path of decision making during the execution of this algorithm, making it easier to understand how a resulting classification is arrived at. Decision trees furthermore inherently highlight the importance of the various features such that they can be employed for feature selection and data interpretation. Their utility for various classification tasks for modeling complex, non-linear relationships between input features and the target variables further enhances their usefulness.

Support Vector Machines (SVMs) [21] are particularly effective for classifications in high-dimensional spaces, making them well suited for datasets with a large number of features. SVMs use kernel functions to handle both linear and non-linear relationships, enabling the use of flexible decision boundaries. By focusing on maximizing the margin between classes, SVMs often achieve strong generalizations and reliable performance on unseen data.

Random Forest (RF) [22] is an ensemble learning algorithm that builds multiple decision trees and combines their predictions for accuracy improvement and overfitting reduction. By aggregating results through majority voting (for classification) or averaging (for regression), RF produces more stable and accurate predictions compared to a single decision tree. It is particularly adept at managing high-dimensional datasets and mitigating noise, as it trains on random subsets of data and features. This diversity helps capture a broad range of data patterns while minimizing the impact of outliers. Random Forest also offers built-in mechanisms for estimating feature importance, which can guide variable selection and provide deeper insights into the data. Moreover, its architecture allows for the parallel training of individual trees, significantly improving computational efficiency on large datasets.

Extreme Gradient Boosting (XGBoost or XGB) [23] is a powerful ensemble method based on gradient boosting. It constructs decision trees in a sequential manner, where each new decision tree attempts to correct the errors made by the previous ones. This iterative refinement leads to high predictive accuracy. XGBoost incorporates several advanced features, such as regularization for overfitting reduction, tree pruning, and efficient missing value handling. It is known for its scalability and speed, often outperforming other algorithms used for structured data problems, and it is widely used in machine learning competitions and real-world applications.

These machine learning techniques have been implemented using Python classifier libraries. To ensure robust model evaluation, stratified k-fold cross-validation has been used, preserving the class distribution across training and validation splits. A modular pipeline architecture has been adopted, enabling easy experimentation with different models and facilitating the comparative analysis of their performance. The following subsections provide detailed descriptions of each of the components of the proposed methodology.

#### Dataset

The study used tissue microarray (TMA) data provided by pathologists at Alberta Precision Laboratory. The dataset includes 1119 samples classified as benign prostate cancer, and it contains clinical and gene expression information. It includes patients diagnosed with prostate cancer who underwent transurethral resection of the prostate (TURP) as part of their treatment. Individuals who had received treatment prior to TURP and exhibited advanced local disease with urinary obstruction while on androgen deprivation therapy (ADT) were classified as having castrate-resistant prostate cancer (CRPC). Tissue samples from these patients were organized into a tissue microarray (TMA). A pathologist involved in the study confirmed the histological diagnosis of each core within the TMA. Each marker would have one slide for TMA and there were 4 TMAs per marker. Grade groups (GGs) were assigned based on the 2018 World Health Organization (WHO) and International Society of Urological Pathology (ISUP) grading criteria by the study’s pathologist (TAB). Clinical information and patient outcomes were collected from the Alberta Cancer registries. The study received ethical approval from the University of Calgary Cumming School of Medicine Ethics Review Board, following the principles of the 1964 Helsinki Declaration and its subsequent revisions or equivalent ethical guidelines (REB Certification #HRE-BA.CC-16-0551_MOD8). Access to the dataset is available to the public upon request to the authors. The list of clinical and gene expression data is presented in Table 3.

### 2.2. Immunohistochemistry (IHC) Procedure

Patient samples were assembled into tissue microarrays (TMAs), and 4 µm-thick formalin-fixed paraffin-embedded (FFPE) sections were prepared and placed on glass slides. Immunohistochemistry (IHC) was then carried out using the Dako Omnis auto-stainer (Agilent, Santa Clara, CA, USA) at the Anatomic Pathology Research Laboratory, which is part of Alberta Precision Research Labs (APRL), following standard protocols. Briefly, the slides were hydrated and antigen retrieval was performed using a Tris-based high pH (9.0) epitope retrieval buffer. The slides were then incubated with a Santa Cruz mouse monoclonal antibody (sc-28383), diluted 1:100 in Dako antibody diluent. Primary and secondary antibodies were incubated for 30 min. After primary antibody incubation, a mouse poly-linker (Agilent) was applied. Detection was performed using the DAB+ Substrate Chromogen system (Agilent), followed by counterstaining with hematoxylin for nuclear visualization. Slides were then dehydrated through graded ethanol and xylene series and coverslipped using Flo-TEXX mounting medium (Lerner Laboratories, Pittsburgh, PA, USA).

IHC was also used to assess the expression of PTEN and ERG. PTEN was classified as either retained (normal) or lost (indicative of functional loss). ERG status was reported as positive or negative, reflecting the presence or absence of ERG gene rearrangement.

#### 2.2.1. Data Preparation

The data were derived from immunohistochemical tests conducted in the laboratory and are logged on spreadsheets according to patient ID. In some cases there were multiple samples taken from different parts of the tumor or from different tumors for a given patient, resulting in multiple rows of data with the same ID but with different test results. This meant that the spreadsheets could not be used directly for input to the machine learning pipelines. Pre-processing techniques were therefor applied, which involved formatting, cleaning, and merging clinical and gene expression data so that the data were compatible with the machine learning pipeline without altering the original information.

##### Data Formatting

Some variations in the gene data made it difficult for it to be interpreted by a machine learning model. To address this, the data were standardized so that all patient data had the same format. This involved simplifying the inputs into categories based on gene signatures.

##### Data Cleaning

Although there were 1119 occurrences of gene expression data, 16 occurrences were excluded because they lacked specific tissue-type labels such as benign prostate cancer (PCa) or neuroendocrine genomic prostate cancer (NGPCA). The 16 samples removed did not affect the representativeness of the data, as we focused only on PCa-related data for risk stratification. The data removed were not labeled appropriately to be categorized into any class, making them redundant. Furthermore, for this research, the focus was solely on PCa-labeled data, disregarding benign data (N=135). There were 1079 instances of usable clinical data, and these instances were used to merge the data in the subsequent pre-processing phase.

##### Data Merging

Data collectors assigned a different identifier to each patient, which aligned with the corresponding number in the clinical data. To combine these datasets, the data were matched using these unique patient numbers. The merging process involved utilizing the structured query language’s (SQL) inner join feature to link the data based on the unique identifiers. Following the merger, there were 939 sets of unique patient data ready for the subsequent stages of processing. Table 4 shows the effect of the data preparation steps (formatting, cleaning, and merging) in terms of quantity. Some of the instances that do not have related clinical data were eliminated.

##### Handling Missing Values

The data still exhibited missing values and an uneven distribution of risk levels despite the pre-processing steps described earlier. Therefore, traditional techniques were applied to handle both the missing data and the class imbalance issues. Missing value imputation is essential as it maintains data integrity, enhances statistical power, improves model performance by enabling data utilization, preserves relationships within the dataset, and ensures comprehensive and reliable interpretations, safeguarding against the loss of valuable information in the analyses. The reason for this is that if instances/subjects with missing values are deleted, the data will be robust, but a large amount of the data might be lost. And, since most of the data are categorical, handling missing values with traditional techniques would be costly. The data were therefore synthesized separately from the models. Traditional ways of handling missing values in a dataset were applied. These include filling up the dataset with either the mean, mode, median, or a constant value. Each of these methods were applied, and the results were compared to choose the appropriate “mode” technique for further study.

##### SMOTE-Tomek Links for Handling the Class Imbalance Problem

Alongside the issue of missing values, the combined dataset also exhibited class imbalance. To address this, the Synthetic Minority Oversampling Technique (SMOTE) [24] was utilized in conjunction with the Tomek link approach [25]. While SMOTE is an oversampling strategy, the Tomek Link serves as an undersampling technique. By integrating these two methods, the resulting dataset benefited from the strengths of both, producing a more balanced and suitable set for further analysis. Unlike simply replicating samples from the minority class, SMOTE, as described by Chawla et al. [24], generates new synthetic examples considering the Euclidean distance to the nearest neighbors within the minority class. The Tomek link, or the T link [25], refines the condensed nearest-neighbor method. Whereas condensed nearest neighbors randomly select k-nearest neighbors from the majority class for removal, the Tomek Link applies specific criteria when selecting pairs for removal. According to these rules, if $d\left(x_{i},x_{k}\right)<d\left(x_{i},x_{j}\right)$ or $d\left(x_{j},x_{k}\right)<d\left(x_{i},x_{j}\right)$ holds, then the pair $\left(x_{i},x_{j}\right)$ is identified as a Tomek link. The quantitative results of the effect of the SMOTE-Tomek Link techniques are shown in the following Table 5.

#### 2.2.2. Applying Machine Learning Methods

The experiments implemented machine learning algorithms, including Decision Tree (DT) [20] and Support Vector Machine (SVM) [21], discussed in the Proposed Methodology section. In addition, ensemble approaches such as Random Forest (RF) [22] and XGBoost (XGB) [23] were used. These models were implemented using the corresponding Python classifier libraries. To ensure a robust evaluation, k-fold stratified cross-validation was applied for all of the methods. The use of a pipeline architecture facilitated seamless integration and experimentation with different models, allowing a comprehensive assessment of their performance metrics. Finally, the results of the various machine learning techniques were compared in order to identify the most effective approach.

For the overall implementation of the methods, Python 3.13.5 packages were used. In addition to the *pandas* [26], *NumPy* [27], *Matplotlib* [28], *re (regular expression)* [29] to manipulate the data frames, and python *scikit-learn* package [30] were used to implement the machine learning methods. The hyperparameters used for the ML methods are provided in Table 6.

To minimize potential batch effects between the training dataset and the independent test dataset, we applied batch effect correction strategies prior model training. Specifically, the data were normalized using z-score standardization. In addition, we have applied the ComBat algorithm [31] from the pyComBat 0.20 package [32], which adjusts for batch-specific variation while preserving biological signals. This ensured that differences in the distribution between batches did not bias biomarker identification or classification outcomes. After correction, we visually inspected the data using principal component analysis (PCA) [33] and confirmed that samples clustered primarily by biological groups (severity levels) rather than by dataset origin. This indicates that batch-related confounding was effectively minimized. Our classification performance on the independent dataset therefore reflects a true biological signal rather than technical artifacts.

The codes used to produce the performance metrics may be accessed using the following link https://github.com/AhmedAlMarouf/PCa_ML_Project (accessed on 17 September 2025).

#### 2.2.3. External Validation

External validation is an extremely important step in biomarker discovery studies. For our case, we used the Human Protein Atlas—Prostate Cancer Proteome dataset [34,35], which contains IHC scoring and expressions for approximately 159 prognostic genes (including PTEN, ERG-related proteins) in 494 prostate adenocarcinoma samples from the Cancer Genome Atlas (TCGA) [36]. One of the main reasons for using this dataset as external validation is the similarity between the dataset that we have utilized. The datasets are similar in nature, since both contain tissue microarray data for a specific list of gene signatures and mapped clinical data. Hence, apart from using the k-fold cross-validation method used in this study, this dataset provides a better understanding of the models when added as in the external validation process.

## 3. Results

This section presents the experimental findings along with an analysis and interpretation of the results. Based on the contribution of this research aiming to find high-aggressive and low-aggressive signatures, four separate experiments were designed. These experiments aimed to predict or classify GGGs, hence addressing the multiclass prediction problem. Each of the experiments included four ML models with a different experimental setup of features or gene signatures.

Experiment 1: High- and low-aggressive signatures were used to classify the Gleason grading groups.Experiment 2: Only high-aggressive signatures were used to classify the Gleason grade groups.Experiment 3: Only low-aggressive signatures were used to classify the Gleason grade groups.Experiment 4: Leave-One-Out Cross-Validation (LOOCV) was used for Gleason grading group prediction.

In this study, the classification of signatures into high- and low-aggressive categories was based on their statistical association with clinical outcomes, particularly overall survival and tumor grade. Specifically, the hazard ratio (HR) was computed for each signature using the univariate Cox proportional hazards models. Signatures with HR > 1.5 and statistically significant *p*-values (*p* < 0.05) were categorized as “high-aggressive”, indicating a stronger association with poor prognosis. Conversely, signatures with HR < 0.75 and *p* < 0.05 were considered “low-aggressive”, implying a potential protective or favorable prognostic effect. The high-aggressive signatures are ERG, LEF1, FRP3, Dynamin, SLC12A2, SLC39A2, AR, Syntenin, SPRK1, EMC2, ARPC1B, SRRT, and CPSF4. The low-aggressive signatures selected by the process are PTEN, ANX4, CHD5, AMACR, Ankyrin-Membrane, ATM, Meis2 (Cyt & Nuc), LAMTOR (Cyt), and BAP1.

Experiments 1, 2, and 3 have been designed to understand how low- and high-aggressive signatures impact overall risk levels, whereas experiment 4 has been designed to understand each of the Gleason grade group predictions. This is why the first three experiments used cross-validation; on the other hand, the last experiment was performed using leave-one-out cross-validation (LOOCV).

The further discussion of the results has been divided into several parts as follows: feature–feature correlation plotting for the biomarkers, comparing the performances of the applied machine learning methods for each experiment, and reporting the importance values of the characteristics of the biomarkers for each experiment. In addition, each of the Gleason grading groups have been compared and applied using the same ML classifiers to find the performance indicators.

### 3.1. Feature–Feature Correlation Plotting for the Biomarkers

Figure 2 displays the correlation values between the features, calculated using the Pearson correlation coefficient (PCC) for all attributes in the dataset. The resulting correlations are visually represented using a color scale ranging from −1 to +1. The figure shows that the features (signatures) are meaningfully structured, with some degree of correlation that may inform grouping or dimensionality reduction, which can be useful in preparing the data for machine learning models. In Figure 2, the observed correlations were quantitatively evaluated using Pearson’s correlation coefficient (PCC). We considered correlations with |r|≥0.60 as strong and those with |r|≥0.80 as very strong, following the established guidelines. Thus, the reported correlations fall within the strong to very strong range, supporting the robustness of the observed associations.

### 3.2. Performance Comparison of Applied Machine Learning Methods

For each of the experiments, the traditional ML classifiers were applied, namely, SVM, DT, RF, and XGB. The performance of the machine learning methods was then assessed using precision, recall, f-measure, and accuracy as the evaluation metrics. The ML pipeline facilitated the identification of the optimal accuracy, as well as other key metrics such as the F1 score. These parameters were computed according to the following equations, and the terms used in each equation are explained in Table 7.(1)$$Precision=\frac{TP}{TP+FP}$$(2)$$Recall=\frac{TP}{TP+FN}$$(3)$$F1-score=\frac{2(Precision\times Recall)}{\left(Precision+Recall\right)}$$(4)$$Accuracy=\frac{TP+TN}{TP+TN+FP+FN}$$

#### 3.2.1. Results of Experiment 1

In experiment 1, a combination of high- and low-aggressive signatures was used as input for the machine learning models. Performance metrics—including precision, recall, F1 score, and accuracy—for the different ML methods, as applied in this experiment, are summarized using a bar graph format in Figure 3. The lowest accuracy (90.22%) was obtained by DT. SVM obtained 93.40% and RF obtained 95.67% accuracy, whereas the highest accuracy (96.85%) was obtained using the XGBoost (Python XGBoost version 3.0.5) method. In addition to achieving the highest accuracy, XGBoost also recorded the top F1 score (96.10%). Although XGBoost achieved the highest accuracy, we evaluated the possibility of overfitting. To minimize this risk, we employed cross-validation (k-fold = 10) and monitored both training and validation performance. The comparable results between training and validation accuracies indicated that the model did not exhibit significant overfitting. Furthermore, hyperparameters such as learning rate (η), max_depth, and regularization parameters (λ, α) were tuned to avoid overfitting, as recommended in prior studies. To provide statistical robustness, we computed 95% confidence intervals for the classification accuracies using bootstrapping (*n* = 1000 resamples). The XGBoost accuracy of 96.85% had a 95% of confidence interval. This interval demonstrates the reliability of the reported performance.

The Decision Tree (DT) method yielded the lowest F1 score at 87.9%, whereas Support Vector Machine (SVM) and Random Forest (RF) demonstrated solid performance with F1 scores of 92% and 94%, respectively.

Figure 4 illustrates the feature importance ranking for experiment 1. As XGBoost outperformed the other ML models in terms of performance, the feature importance values were calculated and ranked using the XGBoost model. The highest importance value (0.175) was obtained by the ERG gene, followed by PTEN, LEF1, ANXN4, CHD5, AMACR, etc. The lowest importance value (0.021) was obtained by CPSF4.

#### 3.2.2. Results of Experiment 2

In experiment 2, only high-aggressive signatures for the ML models were used. Figure 5 shows the performance metrics, including precision, recall, F1 score, and accuracy for experiment 2, considering only the high-aggressive signatures. The comparison between the metrics can be seen by the bars in the figure. The comparison shows that the XGBoost method obtained the highest metrics among the ML classifiers.

Figure 6 illustrates the feature importance ranking for experiment 2, using the XGBoost model as it obtained highest accuracy in this experiment. The highest importance value (0.150) was obtained by the ERG gene, followed by LEF1, FRP3, Dynamin, etc. The lowest importance value (0.041) was obtained by CPSF4.

XGBoost has also obtained similar results in experiment 3, as shown in Figure 7. Among the four ML classifiers, the lowest-performing classifier is DT, while in all three of the first experiments, XGBoost was found to perform better than the other classifiers.

#### 3.2.3. Results of Experiment 3

In experiment 3, only low-aggressive signatures for the ML models were used. Figure 7 therefore shows the performance metrics including precision, recall, F1 score, and accuracy for experiment 3, for these signatures. The comparison between the metrics can be seen in the bar chart. Similarly to experiment 2, the comparison shows that the XGBoost method also obtained the highest metrics compared with the other applied ML classifiers in this experiment. Among the four ML classifiers, the least performing classifier was again DT, while XGBoost was found to perform better than the other classifiers, as was also the case for the first two experiments.

Figure 8 illustrates the feature importance ranking for experiment 3, using the XGBoost model as it obtained highest accuracy in this experiment. The highest importance value (0.150) was obtained by the PTEN gene, followed by ANXN4, CHD5, AMACR, etc., and the lowest importance value (0.049) was obtained by BAP1.

#### 3.2.4. Results of Experiment 4

Table 8 presents a comprehensive comparison of the performance of four machine learning models—Support Vector Machine (SVM), Decision Tree (DT), Random Forest (RF), and XGBoost—across the following five severity levels: Low, Intermediate-Low, Intermediate, Intermediate-High, and High. In Table 9, we show a comparison of the other three models with XGBoost based on accuracy values and *p*-value is mentioned. The *p*-values (Table 9) indicate that XGBoost’s improvement over SVM and DT is statistically significant (p<0.01), while the difference compared to Random Forest is also significant (p=0.03). These results confirm that the superior accuracy of XGBoost is not due to random variation.

Among the models, XGBoost consistently outperformed the others across all severity levels and evaluation metrics. It achieved the highest accuracy, with values exceeding 95% in all categories, and maintained superior precision, recall, and F1-scores. For example, in the High-Severity category, XGBoost attained an accuracy of 95.43%, 94.5% precision, 92.2% recall, and an F1-score of 93.3%. Similarly, strong performance was observed across the remaining severity levels, indicating both robustness and reliability for classification tasks.

In contrast, the Random Forest model demonstrated variable performance. Although it performed well in the High-Severity category (accuracy of 91.64% and F1-score of 90.8%), its accuracy dropped to 80.70% in the Low-Severity group, and the corresponding F1-score was 78.9%. This suggests that RF is more effective at identifying higher severity cases than lower ones. The Decision Tree classifier showed moderate performance, with accuracies ranging from 82.18% to 89.42% across the categories. Its precision, recall, and F1-scores was generally lower than those of XGBoost, indicating a tendency toward less consistent classification, particularly in the Intermediate group where the F1-score dips to 0.777. The Support Vector Machine model also exhibited moderate results, achieving its highest accuracy in the Intermediate category (91.64%) but lower values in the Low and High categories (84.49% and 83.11%, respectively). The F1-scores followed a similar trend, with the highest value of 88.4% in the Intermediate group and the lowest value of 79.8% in the High group.

Overall, XGBoost demonstrated the most balanced and superior performance across all severity levels and metrics, making it the most suitable model for this classification task. The results highlight the advantage of ensemble and boosting techniques, particularly in handling complex, multiclass classification problems within biomedical datasets.

#### 3.2.5. Results of External Validation

The external dataset of TCGA has been utilized to compare the validation with the original dataset. Although we have applied 10-fold cross-validation to produce all the performance metrics in the four above-mentioned experiments, external validation provides stronger foundation and greater acceptability for the model. From Table 8, it is evident that XGBoost provides the highest accuracy in all five classes when compared using the LOOCV method. For external validation, we have used the same XGBoost method and compared them, as shown in Figure 9. The model’s performance was evaluated on both the Original Microarray Dataset (OMD) and an External Validation Dataset (EVD) across five severity levels as follows: Low, Intermediate-Low, Intermediate, Intermediate-High, and High. Overall, the results demonstrate reasonable generalizability, with some variation observed between the datasets.

The model demonstrates strong and generally consistent performance across all severity levels when evaluated on an external validation dataset (EVD) compared to the original microarray dataset (OMD). For Low severity, there was a slight decrease in accuracy (95.12% to 93.10%), precision (93.50% to 91.20%), and recall (88.10% to 85.70%), leading to a modest reduction in the F1 score, possibly due to distributional shifts. In the Intermediate-Low category, precision and accuracy decreased marginally, while recall improved slightly, resulting in a stable F1 score, indicating the balanced performance of the model under moderate variation. Interestingly, for Intermediate severity, although precision decreased slightly (95.51% to 91.85%), precision, recall, and F1 score improved, suggesting strong generalization to unseen data. For Intermediate-High severity, the model showed an increased precision (90.20% to 93.30%) and a slightly lower recall (94.40% to 90.10%), reflecting a trade-off that still maintained a high overall performance. Finally, in the High-Severity class, the model retained high precision and accuracy, while recall improved significantly (92.20% to 95.30%), leading to a higher F1 score (92.30% to 93.40%). These results collectively indicate that the model generalizes well, particularly in more severe cases, and it maintains reliable predictive performance across external data.

## 4. Discussion

This section presents a comprehensive analysis of the results, including a discussion of key gene signatures that demonstrated a strong predictive capacity to determine the severity levels.

### 4.1. Discussion of Results

Ensuring that the data are properly prepared is crucial for the success of any machine learning pipeline. In this study, the initial dataset was not adequately formatted or cleaned, which required data formatting, cleaning, and merging before inputting the data into the machine learning pipeline. In addition, appropriate techniques were applied to address the missing values and class imbalance. These pre-processing steps were essential components of the machine learning workflow.

Figure 2 presents the Pearson correlation values, illustrating the relationships between clinical and signature features. The results indicate satisfactory correlations overall. The most significant negative correlation (−0.5) was observed between the intensity and the dual values of PTEN. Other negative or sub-zero correlation values are associated with variations in the categorical values of the gene signature features.

The Results section provides a clear comparison of six machine learning classifiers—SVM, DT, RF, and XGBoost—to classify severity levels. The findings demonstrate that the ensemble approach of XGBoost offers a significant advantage over the other models, which showed comparatively lower performance. XGBoost achieved the highest overall accuracy at 95.43%, with RF following at 91.64%. Due to its superior performance metrics, XGBoost was selected for further comparative analysis. This aligns with broader research showing that ensemble methods like XGBoost and Random Forest frequently outperform traditional classifiers such as SVM and DT in terms of accuracy and predictive power.

While testing with the external validation dataset, across all severity levels, the model trained on the OMD performs consistently well on the EVD, with only marginal performance degradation in most metrics. In fact, for several severity levels (Intermediate and High), the model outperforms or matches its original performance on external data, indicating strong generalizability and robustness.

These findings validate the clinical or real-world applicability of the model, particularly for more severe cases. Future improvements could focus on improving Low-Severity classification under domain changes, potentially through domain adaptation, data augmentation, or ensemble strategies.

### 4.2. Crosstalk Between PTEN, ERG, and the Related Gene Signatures

Phosphatase and tensin homolog (PTEN) gene mutations are considered a step in the development of many cancer types, including breast cancer, lung cancer, glioblastoma, and prostate cancer. PTEN is a tumor suppressor gene that provides instructions for making proteins that control how quickly cells grow and divide to make new cells. The proteins also restrict the growth of abnormal cells. Many investigations [37,38] have discovered the genomic deletion of PTEN in PCa, and many studies have shown that PTEN is the most commonly lost tumor suppressor gene in PCa [39,40,41]. The clinical implications of this gene were established by [42]. A coexpression analysis was performed between the PTEN-ERG gene and the PTEN-AMACR gene using cBioPortal (https://www.cbioportal.org/) (accessed on 17 September 2025). The TCGA PanCancer Atlas dataset [39] which has 494 samples was used as data to find genes in mRNA expression that correlate with PTEN. The results are shown in Figure 10 and Figure 11. The blue dot in the figures represent each patient having mRNA expression for ERG, PTEN, and AMACR gene signatures.

The figures show that ERG has a lower correlation with PTEN while AMACR has a higher correlation. ERG was chosen because it has higher feature importance in the XML prediction rates and AMACR has lower feature importance in the XML prediction rates.

Not only does PTEN genomic deletion have a poorer outcome in hormone refractory prostate cancer [43], the expression of the ETS-related proto-oncogene (ERG) also plays a crucial role in the ocnogenic mechanism and progression of PCa [44]. The insights gained from the XML interpretations on ERG, PTEN, and other related genes are in line with findings from other studies. Therefore, the method proposed in this paper was able to match its results with existing research, but this was achieved from an explainable machine learning point of view rather than from the point of view of clinical trials.

## 5. Conclusions

In the complex landscape of prostate cancer, integrating machine learning for the identification of biomarkers by severity level marks a significant advancement towards precision medicine. Hence, the identification of robust biomarkers tailored to different severity levels using ML methods heralds a transformative advancement in the diagnosis, prognosis, and treatment of prostate cancer.

The synergy between ML and biomarker identification not only enhances predictive accuracy but also elucidates the underlying biological mechanisms that contribute to disease progression. Through transparent and interpretable models, ML demystifies the intricate web of features that contribute to different severity levels, offering crucial insights for personalized patient care. This approach promotes a deeper understanding of the heterogeneity of the disease, allowing clinicians to differentiate between indolent and aggressive forms of prostate cancer. The elucidation of severity-specific biomarkers holds promise for guiding tailored interventions, optimizing treatment strategies, and minimizing both overdiagnosis and undertreatment.

The identified biomarkers have potential translational relevance in clinical practice. In particular, they could be used as part of diagnostic panels to complement histopathological evaluation (e.g., Gleason classification) and standard serum markers such as PSA. For example, patients with borderline PSA values could benefit from an additional biomarker test to reduce unnecessary biopsies. Moreover, the biomarkers may serve as risk stratification tools, enabling the classification of patients into low-, intermediate-, and high-risk categories, thereby informing treatment decisions such as active surveillance versus definitive therapy. While our findings highlight the biological relevance of genes such as PTEN and ERG, the results remain computational and lack direct clinical validation. Therefore, the clinical implications discussed in this work should be interpreted as hypothesis-generating rather than confirmatory. Future studies with prospective clinical cohorts and functional validation are essential to translate these insights into clinical practice.

In a diagnostic context, these biomarkers could be implemented in multiplex assays (e.g., PCR-based panels, immunohistochemistry, or NGS-based tests) to provide rapid and reproducible results. Integration with existing clinical guidelines (e.g., NCCN [16], EAU [19], and AUA [17,18]) could facilitate their adoption as part of a personalized medicine approach. While additional validation in larger, independent cohorts is required before clinical translation, our findings provide an evidence-based foundation for developing biomarker-driven diagnostic and prognostic assays that may ultimately improve patient outcomes.

## Figures and Tables

**Figure 1 biomedicines-13-02350-f001:**
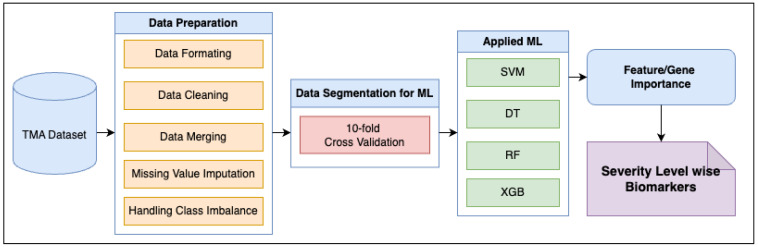
Proposed methodology for severity level-wise biomarker identification.

**Figure 2 biomedicines-13-02350-f002:**
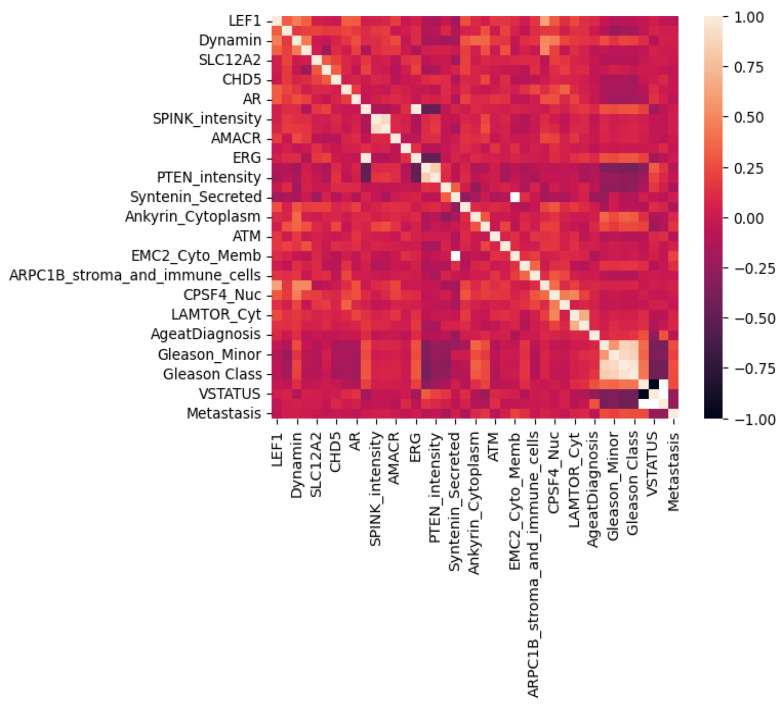
Pearson correlation plot among the clinical and gene expression features.

**Figure 3 biomedicines-13-02350-f003:**
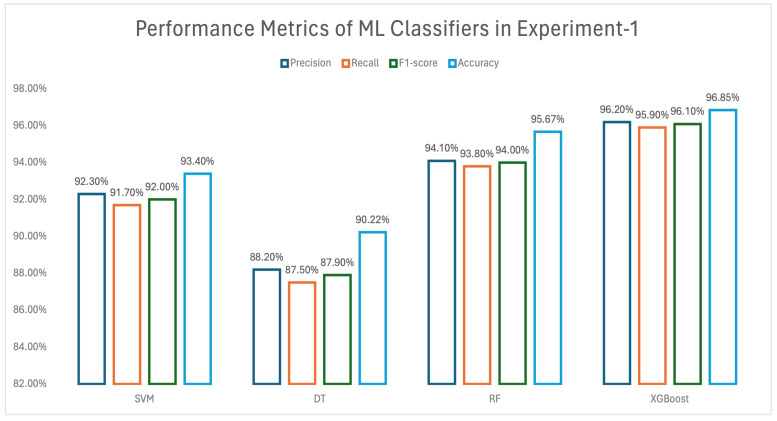
Performance metrics of experiment 1.

**Figure 4 biomedicines-13-02350-f004:**
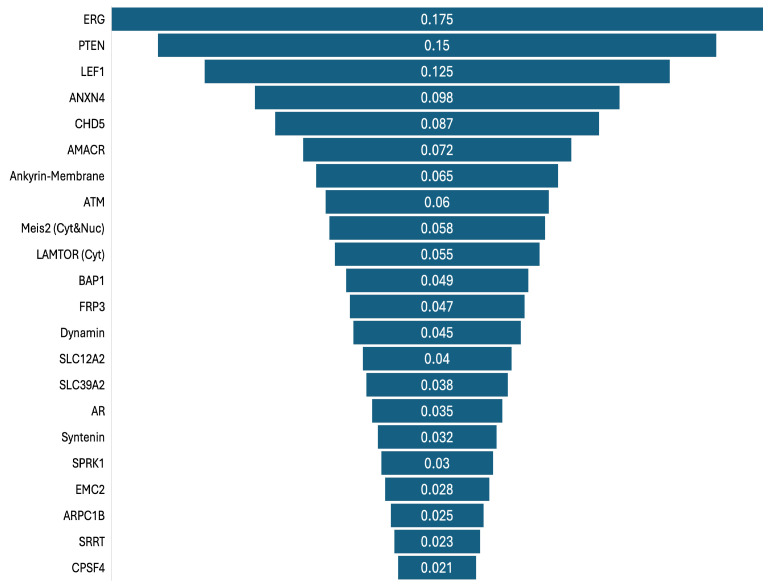
Feature importance ranks of high- and low-aggressive signatures in experiment 1.

**Figure 5 biomedicines-13-02350-f005:**
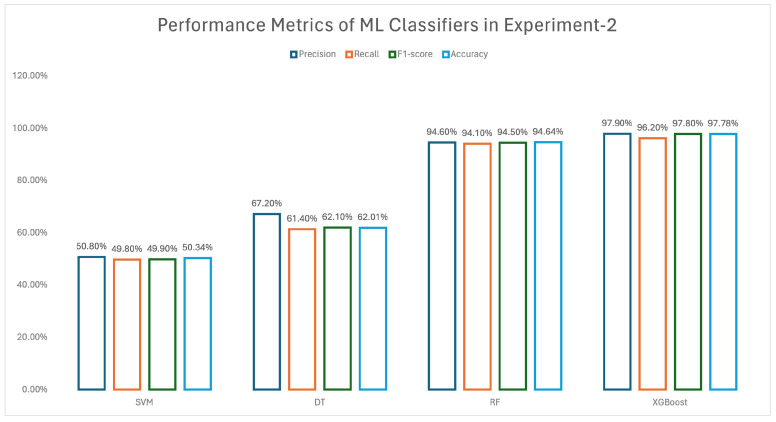
Performance metrics of experiment 2.

**Figure 6 biomedicines-13-02350-f006:**
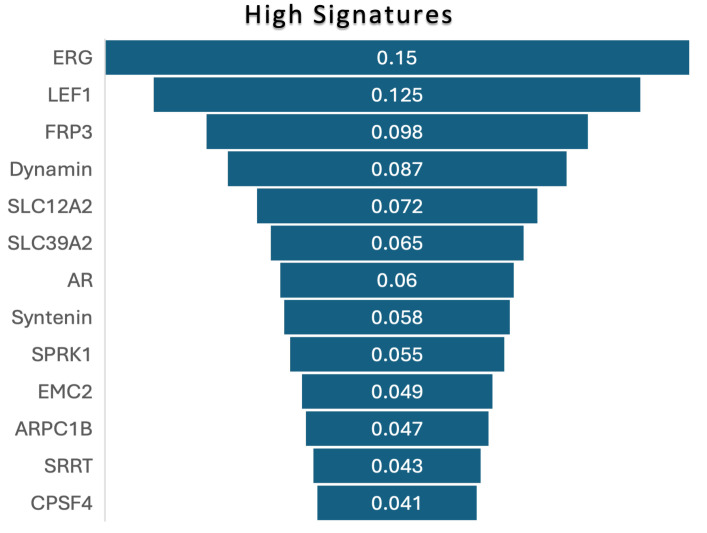
Feature importance ranks of high-aggressive signatures in experiment 2.

**Figure 7 biomedicines-13-02350-f007:**
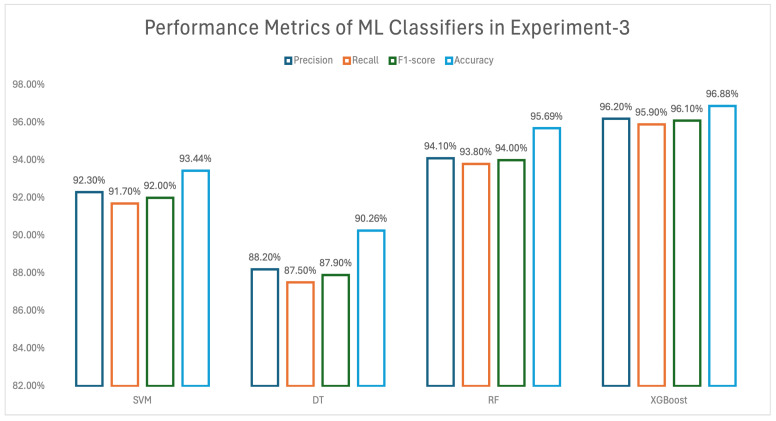
Performance metrics of experiment 3.

**Figure 8 biomedicines-13-02350-f008:**
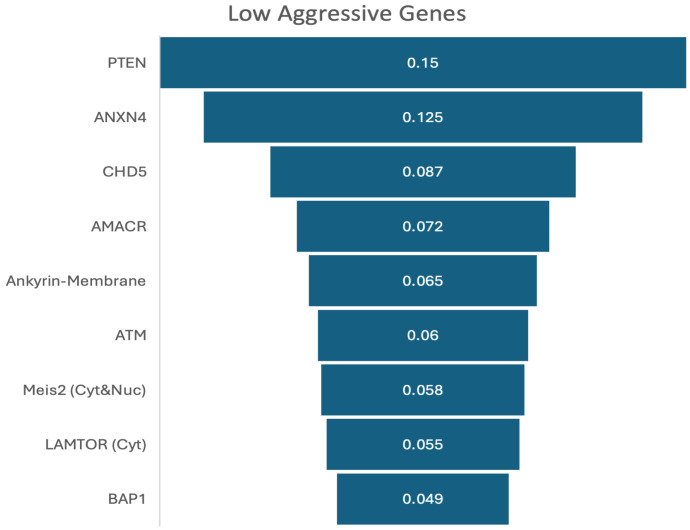
Feature importance ranks of low-aggressive signatures in experiment 3.

**Figure 9 biomedicines-13-02350-f009:**
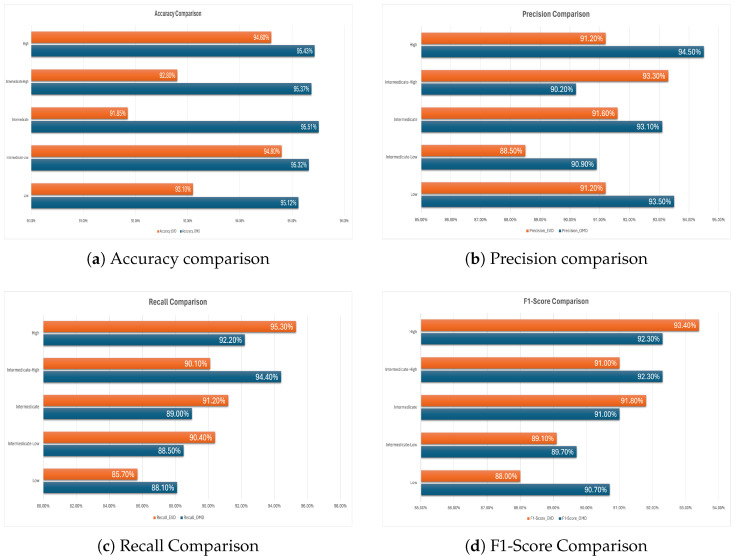
XGBoost performance metric comparison between the original microarray dataset (OMD) and external validation dataset (EVD).

**Figure 10 biomedicines-13-02350-f010:**
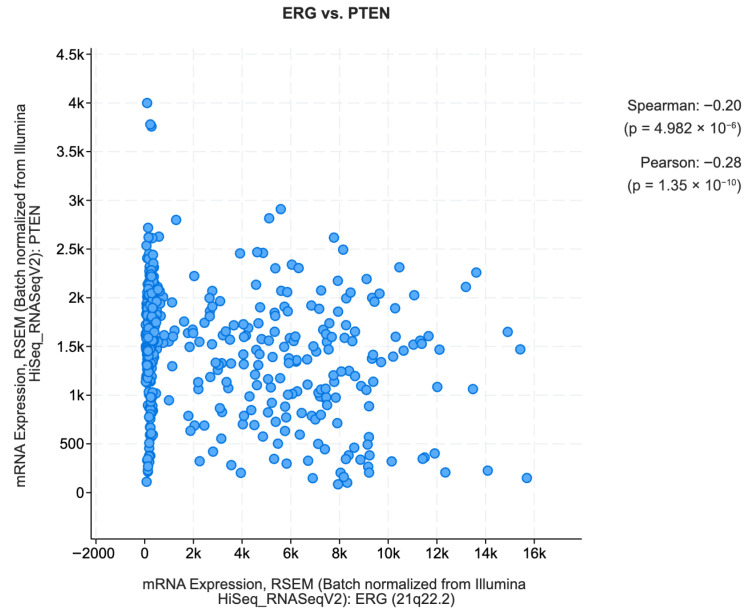
Co-expression plot for PTEN vs. ERG on TCGA-PRAD (batch-normalized from Illumina).

**Figure 11 biomedicines-13-02350-f011:**
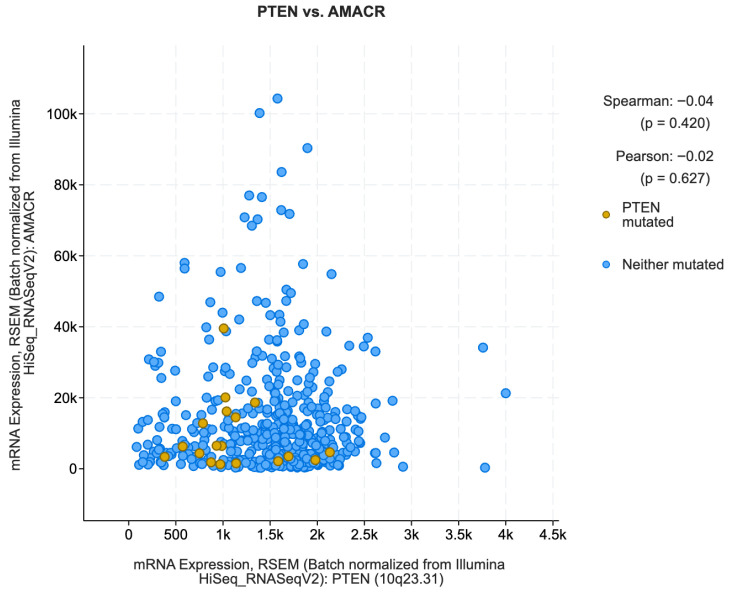
Co-expression plot for PTEN vs. AMACR on TCGA-PRAD (batch-normalized from Illumina).

**Table 1 biomedicines-13-02350-t001:** Summary of representative studies on Gleason score/grade group prediction.

Study (Year)	Modality/Input	Description (Dataset, Methods, Strengths)	Limitations
Arvaniti et al. (2018) [7]	H&E TMA patches	Multi-institutional TMAs; CNN for Gleason pattern classification; reported high quadratic-weighted κ; first robust DL on TMAs with patch-level supervision	Limited to TMAs; domain shift to biopsies
Bulten et al. (2021) [8]	H&E biopsy WSIs	Multi-center European biopsies; end-to-end DL grading; achieved pathologist-level agreement (QWK); strong external validation and clinical applicability	Black-box model; no genomic/clinical integration
Ström et al. (2020) [9]	H&E biopsy WSIs	Population-based registry biopsies (Sweden); DL classifier for cancer detection and grading; high AUC and QWK; prospective diagnostic study design	Regional dataset; limited biomarker insight
Nagpal et al. (2019) [10]	H&E biopsies	Multi-institutional US dataset; DL trained with consensus pathologist labels; pathologist-level QWK; strong reproducibility	Proprietary data/pipeline; limited interpretability
Liu et al. (2019) [11]	mpMRI (T2WI, ADC, DWI)	Institutional cohorts incl. PROSTATEx; radiomics + SVM/RF for high vs. low Gleason; AUC ∼0.75–0.90; non-invasive, pre-biopsy triage potential	Variability across sites/sequences; small validation
Gertych et al. (2015) [12]	H&E patches/ROIs	Institutional TMAs/biopsies; texture/morphology features + SVM/DT; moderate Acc./AUC; interpretable handcrafted features; low data demand	Ceiling performance vs. DL; laborious feature eng.
Klein et al. (2014) [13]	Gene expression panels	Multi-site FFPE cohorts; 17-gene regression-based assay; high AUC for risk stratification; robust clinical SOPs; already clinically deployable	Predicts risk not direct GG; high assay cost

**Table 2 biomedicines-13-02350-t002:** Gleason grade groups [15] with PCa severity levels.

Gleason Grade Group (GGG)	Meaning	Gleason Score (GS)	Severity/Risk Levels
Grade Group-1	The cells look like normal prostate cells. The cancer is likely to grow very slowly, if at all.	GS ≤ 6	Low
Grade Group-2	Most cells still look like normal prostate cells. Cancer is likely to grow slowly.	GS = 7 (3 + 4)	Intermediate-Low
Grade Group-3	The cells look less like normal prostate cells. Cancer is likely to grow at a moderate rate.	GS = 7 (4 + 3)	Intermediate
Grade Group-4	Some cells look abnormal. The cancer might grow quickly or at a moderate rate.	GS = 8	Intermediate-High
Grade Group-5	The cells look to be very abnormal. The cancer is likely to grow quickly.	GS = 9 (4 + 5 or 5 + 4) or GS = 10 (5 + 5)	High

**Table 3 biomedicines-13-02350-t003:** Summary of clinical and gene expression data fields in the dataset.

Clinical Data	Gene Expression Data
Unique ID Number, Age at Diagnosis, Vital Status (Dead/Alive), Age at Death (if dead), Cause of Death, Time Difference (days), Metastasis (Yes/No), Metastasis Sites, Major Gleason Score, Minor Gleason Score, Total Gleason Score	LEF1, FRP3, Dynamin, ANXN4, SLC12A2, SLC39A2, CHD5, AR, ERG (dual), SPINK Intensity, SPINK-1, AMACR, CTK, PTEN (dual), PTEN Intensity, Syntenin Cytoplasm Scoring, Syntenin Secreted, Ankyrin Membrane, Ankyrin Cytoplasm, SPRK1, ATM, MEIS2, EMC2, ARPC1B Tumor Nucleus Cytoplasm, ARPC1B Stromal and Immune Cells, SRRT, CPSF4 Nucleus, CPS4 SRW, LAMTOR Cytoplasm, LAMTOR4 SRW

**Table 4 biomedicines-13-02350-t004:** Effect of data preparation in terms of quantity.

Item	Quantity
Initial no. of instances in dataset	1119
Removing unlabeled data	16
Removing benign class instances	135
After merging with clinical data	939
Data instances used in missing value imputation	939

**Table 5 biomedicines-13-02350-t005:** Effect of SMOTE-Tomek Link techniques on class balancing.

Before		After	
Class/Severity Level	Number of Samples	Class/Severity Level	Number of Samples
Low	287	Low	412
Intermediate-Low	70	Intermediate-Low	412
Intermediate	105	Intermediate	412
Intermediate-High	65	Intermediate-High	412
High	412	High	412

**Table 6 biomedicines-13-02350-t006:** Hyperparameters used for training the machine learning models and pre-processing techniques.

Model/Technique	Hyperparameter	Value
DT	Criterion	Gini Impurity
	Max Depth	None (expand until pure)
	Min Samples Split	2
SVM	Kernel	Radial Basis Function (RBF)
	Regularization Parameter (*C*)	1.0
	Kernel Coefficient (γ)	Scale (1/n_features)
	Tolerance	1 × 10^−3^
RF	Number of Trees (Estimators)	100
	Criterion	Gini Impurity
	Max Depth	None
	Min Samples Split	2
XGBoost	Number of Trees (Estimators)	100
	Learning Rate (η)	0.3
	Max Depth	6
	Subsample	1.0
	Column Sample by Tree	1.0
	Regularization (λ)	1
SMOTE-Tomek	Sampling Strategy	Auto (balance minority class)
	Nearest Neighbors (*k*)	5

**Table 7 biomedicines-13-02350-t007:** Explanation terms used for Equations (1)–(5).

Term	Explanation
True Positive (TP)	The risk level is actually at the same level as predicted
True Negative (TN)	The risk level is actually not the level predicted
False Positive (FP)	The risk level exists when it does not
False Negative (FN)	It does not predict the risk level when it is actually at that level
*N*	Total number of observations
*x*	Individual observation
$\bar{x}$	Population mean

**Table 8 biomedicines-13-02350-t008:** Performance comparison of models across five severity levels (experiment 4).

Model	Metric	Low	Intermediate-Low	Intermediate	Intermediate-High	High
SVM	Accuracy	84.49%	83.51%	91.64%	91.39%	83.11%
	Precision	84.20%	80.30%	90.80%	91.20%	81.40%
	Recall	82.60%	79.70%	86.10%	90.00%	78.30%
	F1-Score	83.40%	80.00%	88.40%	90.60%	79.80%
DT	Accuracy	87.18%	89.42%	82.18%	83.66%	86.24%
	Precision	83.00%	85.40%	78.10%	79.10%	84.00%
	Recall	81.30%	86.00%	77.30%	81.40%	80.50%
	F1-Score	82.10%	85.70%	77.70%	80.20%	82.20%
RF	Accuracy	80.70%	87.11%	86.30%	85.28%	91.64%
	Precision	80.00%	82.30%	83.50%	80.90%	90.50%
	Recall	77.90%	84.40%	81.30%	81.70%	91.20%
	F1-Score	78.90%	83.30%	82.40%	81.30%	90.80%
XGBoost	Accuracy	95.12%	95.32%	95.51%	95.37%	95.43%
	Precision	93.50%	90.90%	93.10%	90.20%	94.50%
	Recall	88.10%	88.50%	89.00%	94.40%	92.20%
	F1-Score	90.70%	89.70%	91.00%	92.30%	93.30%

**Table 9 biomedicines-13-02350-t009:** Accuracy comparison of models across five severity levels with *p*-values against XGBoost.

Model	Low	Intermediate-Low	Intermediate	Intermediate-High	High	*p*-Value vs. XGBoost (Accuracy)
SVM	84.49%	83.51%	91.64%	91.39%	83.11%	<0.01
DT	87.18%	89.42%	82.18%	83.66%	86.24%	<0.01
RF	80.70%	87.11%	86.30%	85.28%	91.64%	0.03
XGBoost	95.12%	95.32%	95.51%	95.37%	95.43%	–

## Data Availability

Data are available from the corresponding author and can be shared with anyone upon reasonable request.

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
