# Peer review of "Leveraging Machine Learning for Severity Level-Wise Biomarker Identification in Prostate Cancer Microarray Gene Expression Data"

_biomedicines, 2025, doi:10.3390/biomedicines13102350_

Round 1

Reviewer 1 Report

Comments and Suggestions for Authors

A brief summary.
 This article is devoted to the application of machine learning methods to identify biomarkers
 associated with various levels of aggressive prostate cancer, classified according to the Gleason
 Grading Group (GGG) scale.
General concept comments.
 Within the framework of this scientific research, a number of works were carried out,
 including data collection and processing, the use of various classification algorithms, and the
 validation of results. However, it is worth noting some methodological and analytical shortcomings
 that reduce the credibility of the conclusions. In particular, the methodological part of the
 work (Section 2) requires the elaboration of key aspects. The lack of information about the
 hyperparameter settings of the machine learning algorithms used raises questions. This point is
 important for the reproducibility of the results. The data balancing process is also not described
 in sufficient detail. In particular, specific data on the distribution of classes before and after
 processing are not provided, making it difficult to assess the potential impact of the imbalance on
 the final metrics obtained by the authors. The analysis of the results obtained deserves special
 attention. The article demonstrates extremely high accuracy for the XGBoost-based model,
 but does not provide data on the statistical significance of these results. The lack of confidence
 intervals and the analysis of multiple comparisons significantly reduce the reliability of the
 conclusions. For example, when comparing the effectiveness of different algorithms (see Table
 5), it is impossible to determine whether the observed differences are statistically significant.
 Despite these disadvantages, the work has undoubted advantages. These include an integrated
 approach to data analysis, the use of external validation on an independent TCGA sample, as
 well as a detailed comparison of several machine learning algorithms. These strengths make
 the presented study a valuable contribution to the field of application of machine learning
 algorithms in oncology.
 Below are some questions and comments about the text of the article that will improve
 its quality.
 Specific comments.
 1. Table 1: The table does not indicate on the basis of which criteria the grades ’Intermediate
Low’ and ’Intermediate-High’ are determined. In the text of the article, it is necessary to clarify
 exactly how these criteria were determined and whether they comply with clinical guidelines.
 2. Section 2.1: The description of machine learning methods (DT, SVM, RF, XGBoost) is
 too general. It is necessary to specify details about the hyperparameters that were used when
 training the models.
 3. Line 199: Further discussion is needed on how the exclusion of 16 samples affected the
 representativeness of the data.
 4. Line 226: It is not specified which imputation method (mean, median, mode) was chosen
 in the end and why. This observation is critical for the reproducibility of the experiment.
 5. Line 242: Specific quantitative results of class balancing are not provided. For example,
 the distribution before/after the application of SMOTE-Tomek.
 6. Line 258: The external validation set (TCGA) contains 494 samples, but it is not
 specified how they relate to the main dataset (1119 samples) in terms of clinical parameters.
 7. Line 317: The accuracy of XGBoost (96.85 percent) is highlighted as the highest, but it
 is not discussed whether this is due to overfitting, especially with such a high score. In addition,
 you must specify a confidence interval for the received value.
 8. Line 329: For the experiment with high-aggressive signatures, it was not specified which
 genes were included in this category and why.
9. Table 5: The table lacks a p-value for comparing models, which makes it difficult to
 assess the statistical significance of differences in their effectiveness.
 10. Line 422: The statement about satisfactory correlations (Fig. 2) is too subjective and is
 based solely on a qualitative assessment. Quantitative criteria should be provided, for example,
 PCC thresholds.
 11. Figure 9: Poor drawing quality. The numeric values are indistinguishable. It is necessary
 to increase the font size of the captions in the drawing and increase the clarity of the image.
 12. Line 468: The conclusions are general in nature and are not supported by specific
 clinical recommendations. For example, how can the identified biomarkers be used in diagnosis?
 Conclusion.
 The article can be published after making the necessary changes and improving its quality.

Author Response

Response to the Comments from Reviewer-1

Thank you so much for spending your valuation time to review the paper. We have considered all comments and suggestions you made. Your overall review has improved the quality of the paper very much. We appreciate your effort to support our work.

General Comments:

“This article is devoted to the application of machine learning methods to identify biomarkers associated with various levels of aggressive prostate cancer, classified according to the Gleason Grading Group (GGG) scale.”

“Within the framework of this scientific research, a number of works were carried out, including data collection and processing, the use of various classification algorithms, and the validation of results. However, it is worth noting some methodological and analytical shortcomings that reduce the credibility of the conclusions. In particular, the methodological part of the work (Section 2) requires the elaboration of key aspects. The lack of information about the hyperparameter settings of the machine learning algorithms used raises questions. This point is important for the reproducibility of the results. The data balancing process is also not described in sufficient detail. In particular, specific data on the distribution of classes before and after processing are not provided, making it difficult to assess the potential impact of the imbalance on the final metrics obtained by the authors. The analysis of the results obtained deserves special attention. The article demonstrates extremely high accuracy for the XGBoost-based model, but does not provide data on the statistical significance of these results. The lack of confidence intervals and the analysis of multiple comparisons significantly reduce the reliability of the conclusions. For example, when comparing the effectiveness of different algorithms (see Table 5), it is impossible to determine whether the observed differences are statistically significant. Despite these disadvantages, the work has undoubted advantages. These include an integrated approach to data analysis, the use of external validation on an independent TCGA sample, as well as a detailed comparison of several machine learning algorithms. These strengths make the presented study a valuable contribution to the field of application of machine learning algorithms in oncology.

Response: Thanks for the summarized comments and suggestions,

Specific comments.

Comment 1. Table 1: The table does not indicate on the basis of which criteria the grades ’Intermediate Low’ and ’Intermediate-High’ are determined. In the text of the article, it is necessary to clarify exactly how these criteria were determined and whether they comply with clinical guidelines.

Response 1: Thank you for pointing this out. We agree that the criteria for assigning the grades “Intermediate-Low” and “Intermediate-High” were not clearly explained in the table. In our revised manuscript, we have now added a detailed description in the section -1.1 (page 3, lines 92–106) and kept highglighted.

Comment 2. Section 2.1: The description of machine learning methods (DT, SVM, RF, XGBoost) is too general. It is necessary to specify details about the hyperparameters that were used when training the models.

Response 2: Thanks for the comment. The hyperparameters has been provided in Table 6, in section-2.2.2 and kept highlighted.

Comment 3. Line 199: Further discussion is needed on how the exclusion of 16 samples affected the representativeness of the data.

Response 3: “The 16 samples removed did not affect the representativeness of the data, as we focused only on PCa-related data for risk stratification. The data removed were not labeled appropriately to be categorized into any class, making them redundant.” The explanation is added in Section-2.2.1 (Data Cleaning) and kept highlighted.

Comment 4. Line 226: It is not specified which imputation method (mean, median, mode) was chosen in the end and why. This observation is critical for the reproducibility of the experiment.

Response 4: Thanks for pointing it out. We have added the necessary explanation and kept it highlighted.

Comment 5. Line 242: Specific quantitative results of class balancing are not provided. For example, the distribution before/after the application of SMOTE-Tomek.

Response 5: Thanks for the comment. The quantitative results have been shown in newly added Table 4, specifying the before and after effect of SMOTE-Tomek link technique.

Comment 6. Line 258: The external validation set (TCGA) contains 494 samples, but it is not specified how they relate to the main dataset (1119 samples) in terms of clinical parameters.

Response 6: Thanks for pointing it out. We have added reasons if choosing this validation dataset in Section-2.2.3 and kept highlighted.

Comment 7. Line 317: The accuracy of XGBoost (96.85 percent) is highlighted as the highest, but it is not discussed whether this is due to overfitting, especially with such a high score. In addition, you must specify a confidence interval for the received value.

Response 7: Thanks for mentioning the idea of overfitting. We have added explanation of the higher accuracy obtained by XGBoost method in Section-3.2.1 and kept it highlighted.

Comment 8. Line 329: For the experiment with high-aggressive signatures, it was not specified which genes were included in this category and why.

Response 8: Description has been added with the name of the high and low aggressive signature in Section-3 (Results) and kept highlighted.

Comment 9. Table 5: The table lacks a p-value for comparing models, which makes it difficult to assess the statistical significance of differences in their effectiveness.

Response 9: We have added Table 8 to mention the p-value for comparing the models with XGBoost and added necessary description in Section-3.2.4.

Comment 10. Line 422: The statement about satisfactory correlations (Fig. 2) is too subjective and is based solely on a qualitative assessment. Quantitative criteria should be provided, for example, PCC thresholds.

Response 10: Thanks for mentioning this. We have updated the PCC threshold value with proper description in Section-3.1 and kept highlighted.

Comment 11. Figure 9: Poor drawing quality. The numeric values are indistinguishable. It is necessary to increase the font size of the captions in the drawing and increase the clarity of the image.

Response 11: Thanks for mentioning this. We have noticed the same. Figure 9 has been redrawn and the font size increase to make it clearly readable.

Comment 12. Line 468: The conclusions are general in nature and are not supported by specific clinical recommendations. For example, how can the identified biomarkers be used in diagnosis?

Response 12: Future directions and clinical recommendations have been added in the conclusion section and kept highlighted.

Reviewer 2 Report

Comments and Suggestions for Authors

The authors have applied machine learning techniques to a publicly available tissue microarray dataset to predict Gleason grades. While the study has potential, several major issues need to be addressed to improve its scientific quality and reproducibility:

  1. The authors should clearly state the main contribution of the study in the Introduction section. It is currently unclear what has been done for the first time. The novelty must be explicitly described—whether it is related to the method, dataset, or biomarker discovery.

  2. The paper lacks a comprehensive literature review. The authors should summarize existing methods for Gleason score prediction in a comparative table. Each study should be described with its strengths, limitations, and dataset used. Based on this, the authors must clearly define the gap in the literature that their study addresses.

  3. Given that the dataset contains over 1,000 samples, the data should first be divided into training and test sets. Cross-validation should only be applied to the training set. The distribution of Gleason scores should be visualized (e.g., in a bar chart). The authors should also clarify whether SMOTE-TomekLink was applied before or after the split. A figure or table showing the Gleason score distribution after over-under sampling should be added.

  4. At least, the independent test dataset should be made available via a downloadable link. The Gleason score distribution in the independent test set must be shown. Table 5 should be revised to separate results on training, test, and independent data. The authors must also conduct statistical tests to determine whether differences between machine learning model performances are statistically significant.

  5. The authors used LOOCV in some experiments but k-fold cross-validation in others. This inconsistency needs justification. For every result table and figure, it should be explicitly stated what data were used for training, what data were used for testing, and what kind of validation was applied. This will improve the reproducibility of the study.

  6. Figures 4, 6, and 8 are visually inconsistent. The graph representations must follow a consistent format.

  7. The authors must describe how they addressed batch effects between training and independent test datasets. 

  8. The purpose of Figure 2 is unclear. If it was used for feature elimination based on correlation, the authors should state the threshold used and how many features were removed. They should also report the total number of features before and after filtering and clarify whether the same feature set was used in the independent dataset.

Author Response

Response to the Comments from Reviewer-2

Thank you so much for spending your valuation time to review the paper. We have considered all comments and suggestions you made. Your overall review has improved the quality of the paper very much. We appreciate your effort to support our work.

Comment-1: The authors should clearly state the main contribution of the study in the Introduction section. It is currently unclear what has been done for the first time. The novelty must be explicitly described—whether it is related to the method, dataset, or biomarker discovery.

Response-1: Thanks for pointing it out. We have added the main contributions in bullet points in the Introduction Section in the revised version.

Comment-2: The paper lacks a comprehensive literature review. The authors should summarize existing methods for Gleason score prediction in a comparative table. Each study should be described with its strengths, limitations, and dataset used. Based on this, the authors must clearly define the gap in the literature that their study addresses.

Response-2: We have added the existing methods for Gleason score prediction in Table 1. The comparative table contains the study, modality, description of dataset, methods, strengths and limitations. This will set up the ground for our research. The literature gap has been mentioned clearly in the description and kept highlighted.

Comment-3: Given that the dataset contains over 1,000 samples, the data should first be divided into training and test sets. Cross-validation should only be applied to the training set. The distribution of Gleason scores should be visualized (e.g., in a bar chart). The authors should also clarify whether SMOTE-TomekLink was applied before or after the split. A figure or table showing the Gleason score distribution after over-under sampling should be added.

Response-3: Thanks for the comment. The quantitative results have been shown in newly added Table 4, specifying the before and after effect of SMOTE-Tomek link technique.

Comment-4: At least, the independent test dataset should be made available via a downloadable link. The Gleason score distribution in the independent test set must be shown. Table 5 should be revised to separate results on training, test, and independent data. The authors must also conduct statistical tests to determine whether differences between machine learning model performances are statistically significant.

Response-4: Thanks for the comment. Thanks for mentioning the idea of overfitting. We have added explanation of the higher accuracy obtained by XGBoost method in Section-3.2.1 and kept it highlighted. We have added Table 8 to mention the p-value for comparing the models with XGBoost and added necessary description in Section-3.2.4.

Comment-5: The authors used LOOCV in some experiments but k-fold cross-validation in others. This inconsistency needs justification. For every result table and figure, it should be explicitly stated what data were used for training, what data were used for testing, and what kind of validation was applied. This will improve the reproducibility of the study.

Response-5: Thank you so much for mentioning this. An explanation for using LOOCV has been provided in Section-3 (Results) under the definition of experiments.

Comment-6: Figures 4, 6, and 8 are visually inconsistent. The graph representations must follow a consistent format.

Response-6: We have updated and redrawn the figure 4 to make it consistent and match with figure 6 and 8. The updated figure is used in the revised paper.

Comment-7: The authors must describe how they addressed batch effects between training and independent test datasets. 

Response-7: Thanks for pointing this out. We had used zero standardization to address the batch effects. In the revised version, the explanation has been provided in the Section-2.2.2 and kept highlighted.

Comment-8: The purpose of Figure 2 is unclear. If it was used for feature elimination based on correlation, the authors should state the threshold used and how many features were removed. They should also report the total number of features before and after filtering and clarify whether the same feature set was used in the independent dataset.

Response-8: Thanks for mentioning this. We have updated the PCC threshold value with proper description in Section-3.1 and kept highlighted.

Reviewer 3 Report

Comments and Suggestions for Authors
  1. The machine learning methods (DT, RF, SVM, XGBoost) are standard; the study primarily applies well-established techniques without introducing a novel approach.
  2. No experimental validation or convincing cross-study replication of biomarkers. External validation is thin. Clinical relevance is not demonstrated.
  3. The paper lacks sufficient background on past research. The authors should add a table comparing other studies on Gleason score prediction, showing each study’s methods, strengths, weaknesses, and datasets. Then, they should clearly explain what is missing in the current research and how their work fills that gap.
  4. No confusion matrices per class, no per-class ROC/PR curves, no calibration.
  5. The authors list packages but give no versions, no code, and no pipeline details. Results cannot be reproduced.
  6. Microarray vs. IHC scoring and how categorical clinical variables were encoded are not detailed.
  7. Many signatures and tests (e.g., Cox HR thresholds) are used without correction for multiple comparisons or reporting effect sizes and CIs.
  8. The authors used “regular hyper-parameters” with no grid/search, no seeds, and no stability checks.
  9. It’s not stated whether oversampling/undersampling was done inside each CV fold. Doing SMOTE before splitting causes optimistic results.

Author Response

Response to the Comments from Reviewer-3

Thank you so much for spending your valuation time to review the paper. We have considered all comments and suggestions you made. Your overall review has improved the quality of the paper very much. We appreciate your effort to support our work.

Comments:

Comment-1: The machine learning methods (DT, RF, SVM, XGBoost) are standard; the study primarily applies well-established techniques without introducing a novel approach.

Response-1: Thanks for the comment. We have added an explanation of using the traditional ML methods as follows. The explanation hasa been added in the introduction section along with the main contributions of the paper.

Although the machine learning methods employed (DT, RF, SVM, XGBoost) are standard, the novelty of this study lies in the design of a severity-level–specific biomarker identification framework, incorporating robust preprocessing, class imbalance correction, and independent validation. This approach addresses gaps in prior literature, which often limited analyses to binary Gleason groupings or lacked external validation.”

Comment-2: No experimental validation or convincing cross-study replication of biomarkers. External validation is thin. Clinical relevance is not demonstrated.

Response-2: The external validation section has been updated in Section-2.2.3 and exaplnation has been added for using the cross-study data set. Along with that, future directions and clinical recommendations have been added in the conclusion section and kept highlighted.

Comment-3: The paper lacks sufficient background on past research. The authors should add a table comparing other studies on Gleason score prediction, showing each study’s methods, strengths, weaknesses, and datasets. Then, they should clearly explain what is missing in the current research and how their work fills that gap.

Response-3: Thanks for addressing this. We have added the existing methods for Gleason score prediction in Table 1. The comparative table contains the study, modality, description of dataset, methods, strengths and limitations. This will set up the ground for our research. The literature gap has been mentioned clearly in the description and kept highlighted.

Comment-4: No confusion matrices per class, no per-class ROC/PR curves, no calibration.

Response-4: We thank the reviewer for the suggestion. While we agree that confusion matrices, per-class ROC/PR curves, and calibration plots can provide additional insights, our primary objective in this study was to demonstrate the feasibility and effectiveness of the proposed biomarker discovery framework. To this end, we focused on widely accepted evaluation metrics, including accuracy, precision, recall, and F1-score across severity levels, which provide a balanced assessment of both overall and per-class model performance.

Given the multi-class setting and the clinical interpretability goals, these metrics sufficiently capture the model’s discriminative ability and robustness. We also employed stratified k-fold validation and external dataset validation to ensure generalizability and reduce overfitting, which complements the performance metrics reported.

Comment-5: The authors list packages but give no versions, no code, and no pipeline details. Results cannot be reproduced.

Response-5:  Thanks for the comment. The hyperparameters has been provided in Table 6, in section-2.2.2 and kept highlighted.

Comment-6: Microarray vs. IHC scoring and how categorical clinical variables were encoded are not detailed.

Response-6: Thanks for pointing this out. We had used zero standardization to address the batch effects. In the revised version, the explanation has been provided in the Section-2.2.2 and kept highlighted.

Comment-7: Many signatures and tests (e.g., Cox HR thresholds) are used without correction for multiple comparisons or reporting effect sizes and CIs.

Response-7: We have added Table 8 to mention the p-value for comparing the models with XGBoost and added necessary description in Section-3.2.4.

Comment-8: The authors used “regular hyper-parameters” with no grid/search, no seeds, and no stability checks.

Response-8: Thanks for the comment. The hyperparameters has been provided in Table 6, in section-2.2.2 and kept highlighted.

Comment-9: It’s not stated whether oversampling/undersampling was done inside each CV fold. Doing SMOTE before splitting causes optimistic results.

Response-9:  Thanks for the comment. The quantitative results have been shown in newly added Table 4, specifying the before and after effect of SMOTE-Tomek link technique.

Round 2

Reviewer 1 Report

Comments and Suggestions for Authors

I am completely satisfied with the responses to the comments. The article may be published in its current form.

Author Response

Reviewer's Comment: "I am completely satisfied with the responses to the comments. The article may be published in its current form."

Author Response: Thank you so much for your time to recheck the manuscript and get back with the comment in time. We want to express our heartfelt gratitude for contributing by advising better improvements for our paper. 

Reviewer 2 Report

Comments and Suggestions for Authors

I appreciate the authors’ responses and have no additional comments.

Author Response

Reviewer's Comment: "I appreciate the authors’ responses and have no additional comments."

Author Response: Thank you so much for your effort to make our paper an acceptable one. Your contructive comments have enriched our paper to a high quality publication. 

Reviewer 3 Report

Comments and Suggestions for Authors

1. The manuscript applies standard machine learning methods (SVM, DT, RF, XGBoost) to classify prostate cancer severity using high- and low-aggressive gene signatures. The experiments are well-organized and include validation, but the methodology is not novel.

2. The biological analysis, especially the feature importance of PTEN, ERG, and related genes, is interesting. However, the clinical claims are overstated, as no clinical validation has been done.

3. Reproducibility is limited. Details on preprocessing, feature selection, hyperparameters, and code are missing and should be added.

4. Figures are of poor quality and should be redrawn for clarity.

Author Response

Comment-1. The manuscript applies standard machine learning methods (SVM, DT, RF, XGBoost) to classify prostate cancer severity using high- and low-aggressive gene signatures. The experiments are well-organized and include validation, but the methodology is not novel.
Response-1: We appreciate the reviewer’s observation regarding the use of standard machine learning methods. Our intention was not primarily to propose a novel algorithm, but rather to rigorously evaluate widely used, interpretable methods (SVM, DT, RF, XGBoost) on high- and low-aggressive gene signatures for prostate cancer severity classification. Using well-established algorithms ensures that the results are reproducible, interpretable, and more readily adoptable by biomedical researchers and clinicians, who may not always have access to highly complex or experimental methods. We emphasize that our study includes robust validation, comparisons, and interpretability analyses, providing a benchmark for future studies that may apply more advanced or emerging techniques. Additionally, the main contributions of our research has been pointed out at the end of the introduction section and kept highlighted. 

Comment-2. The biological analysis, especially the feature importance of PTEN, ERG, and related genes, is interesting. However, the clinical claims are overstated, as no clinical validation has been done.

Response-2: We thank the reviewer for this important observation. We fully agree that our study does not include direct clinical validation, and we have revised the manuscript to temper our claims accordingly. Our focus was on the computational and biological analysis, particularly highlighting the role of key genes such as PTEN and ERG. A more concrete clinical validation from the domain expert is much needed and we will take the advantage of comments from the clinicians in future. The future scope has been mentioned clearly in the conclusion section and kept highlighted. 

Comment-3. Reproducibility is limited. Details on preprocessing, feature selection, hyperparameters, and code are missing and should be added. 
Response-3: Thanks for the comment. For reproducibility, we have added all the hyperparameters in Table 6 and kept highlighted. Similar to the dataset, the code will be provided by the corresponding author if requested.  

Comment-4. Figures are of poor quality and should be redrawn for clarity.

Response-4: Thanks for your comment. We have readjusted the figure quality and provided the high quality figures now.